# Intelligent Early Fault Diagnosis of Space Flywheel Rotor System

**DOI:** 10.3390/s23198198

**Published:** 2023-09-30

**Authors:** Hui Liao, Pengfei Xie, Sier Deng, Hengdi Wang

**Affiliations:** 1School of Mechatronics Engineering, Northwestern Polytechnical University, Xi’an 710071, China; 2418227282@mail.nwpu.edu.cn; 2School of Mechanical and Power Engineering, Zhengzhou University, Zhengzhou 450001, China; 3Luoyang Bearing Research Institute Co., Ltd., Luoyang 471039, China; 4School of Mechatronics Engineering, Henan University of Science and Technology, Luoyang 471003, China; 9901432@haust.edu.cn

**Keywords:** space flywheel rotor system, intelligent fault diagnosis, data with insufficient labels, missing fault types, hierarchical branch structure, similarity clustering, multi-channel convolutional neural networks

## Abstract

Three frequently encountered problems—a variety of fault types, data with insufficient labels, and missing fault types—are the common challenges in the early fault diagnosis of space flywheel rotor systems. Focusing on the above issues, this paper proposes an intelligent early fault diagnosis method based on the multi-channel convolutional neural network with hierarchical branch and similarity clustering (HB-SC-MCCNN). First, a similarity clustering (SC) method is integrated into the parameter-shared dual MCCNN architecture to set up as the basic structural block. The hierarchical branch model and additional loss are then added to SC-MCCNN to form a hierarchical branch network, which simplifies the problem of fault multi-classification into binary classification with multi-steps. Based on the self-learning characteristics of the proposed model, the unlabeled data and the missing fault types in the training set are re-labeled to realize the re-training of the network. The results of the experiments for comparing the abilities between the proposed method and several advanced deep learning models confirm that on the established early fault dataset of the space flywheel rotor system, the proposed method successfully achieves the hierarchical diagnosis and presents stronger competitiveness in the case of insufficient labeled data and missing fault types at the same time.

## 1. Introduction

The space flywheel rotor system, as the rotating support core of the attitude control system, is extremely important in satellite equipment. A catastrophic consequence would occur once the flywheel system malfunctions [1,2,3]. Pair preload space ball bearings [4] are generally used in space flywheel rotor systems. This type of bearing uses a porous oil-impregnated cage [5] to realize the internal micro-circulation lubrication [6], and is unable to be maintained for life.

The operational accuracy of the flywheel could be reduced by various factors, such as the abnormal rubbing of cages [7], wear [8], or the surface damage defects [9] of bearings, which could further lead to the failure of the attitude control system. Given the complex types of early faults and the insufficient labeling data and missing fault types in flywheel rotor systems, the intelligent early fault diagnosis of space flywheel rotor systems becomes a challenging task.

Furthermore, the electromechanical rotary system and the space flywheel rotor system are both rotary mechanisms. However, due to the unique operating conditions and precision requirements of the latter, its fault characteristics are not entirely the same as the former. Replicating many fault diagnosis methods from the former to the latter poses several challenges [10,11,12,13,14].

In 2015, Lecun et al. [15] proposed deep learning (DL) in nature. Due to the features of automatic feature learning, powerful pattern classification, and the reduction in the need for prior knowledge and artificial experience, the DL is more intelligent and adaptive. Both the wide application in many fields and the amazing results achieved [16,17,18] reflect the excellence of DL.

Some researchers have designed many intelligent bearing diagnosis methods by introducing DL [19,20,21,22,23], especially the method of rolling bearing fault diagnosis based on CNN. However, these studies generally assume that there are sufficient tag data samples and no missing fault types in the training dataset [24,25]. In practical applications, it is generally difficult or costly to obtain a sufficient number of fault data samples, while unlabeled and normal data samples are relatively easy to obtain. This results in an imbalance in the number of data samples of various types manifested as missing fault types and insufficient labeled data samples [26,27,28,29]. Therefore, the assumptions of those studies are unreasonable and would have a negative impact on the practical applications.

Transfer learning and unsupervised learning are currently the most commonly used methods to address the lack of labeled data. Liao [30] proposed a cross-domain fault diagnosis method based on the dynamic distribution adaptive transfer network (DDATN) to solve the problem of the impact of differences between marginal distribution and conditional distribution on domain divergence. This method utilizes the instance-weighted dynamic maximum mean difference for dynamic distribution adaptation to adapt the target domain to the source domain. Zhang et al. [31,32] monitored tool wear and bearing fault under different operating conditions, and proposed multi-label transfer reinforcement learning (ML-TRL) by integrating the feature extraction capabilities of deep reinforcement learning (DRL) and the knowledge transfer capabilities of transfer learning. Li [33,34] proposed a fault diagnosis method based on deep learning for rotating machinery with a small amount of supervised (labeled) data and sufficient unsupervised data. Through confrontation training between feature extractors and domain discriminators, the problem of domain generalization in fault diagnosis was solved. The reliability of the proposed method was verified through the CWRU rolling bearing dataset and the high-speed train bogie bearing dataset.

In summary, current research on fault diagnosis using transfer learning and unsupervised learning mainly focuses on the issue of insufficient tag data, but there is very little research on the issue of missing fault types in training sets, and the effect of transfer learning is easily affected by the amount of fully labeled source domain data. The core idea of unsupervised learning is to use incomplete data modeling, utilize the self-learning characteristics of the constructed network model, relabel unlabeled data, and retrain the network model. Given this, this paper draws on the core idea of unsupervised learning and focuses on the accuracy of relabeling in the case of insufficient labeled data and missing fault types. Moreover, we propose an intelligent early fault diagnosis method for space flywheel rotor systems based on HB-SC-MCCNN. The main contributions of this paper are as follows:Integrating similarity clustering into a dual MCCNN architecture with shared parameters to achieve accurate fault detection in the event of insufficient labeled data (binary classification, i.e., determining whether there is a fault in the rotor system).Simplifying the multi-classification problem to a multi-step binary classification problem by introducing a hierarchical branch, and accurate bearing fault location (multi-classification) is achieved when labeled data are insufficient.The SC model enables the convolutional network model to self-learn new fault types and realizes the relabeling of missing fault types in the training dataset.

The rest of this paper is organized as follows: we present the issues studied in this paper in Section 2. Section 3 describes the proposed intelligent early fault diagnosis method for space flywheel rotor systems based on HB-SC-MCCNN in detail. Then, in Section 4, an experimental evaluation of the model is conducted using the established early fault dataset of the satellite flywheel rotor system, and the results are discussed. Section 5 summarizes this paper.

## 2. Problem Formulation

Let $D={\left[T_{N},T_{F}\right]}^{n}={\left[T_{N},T_{Fc},T_{Fw},T_{Fi},T_{Fo},T_{Fb}\right]}^{n}$ represent the monitoring dataset (labeled data with the operation status of flywheel rotor system), where *n* represents the number of samples in the dataset, $T_{N}={\left\{\left(T_{N},T_{F}\right)\right\}}_{i=1}^{n_{N}}$ represents the normal dataset of the system, $X_{i}^{N}$ is the normal data sample, $N_{i}$ is the corresponding operation status normal label, and $n_{N}$ is the number of normal data samples. $T_{Fc}={\left\{\left(X_{i}^{Fc},Fc_{i}\right)\right\}}_{i=1}^{n_{F}c}$, $T_{Fw}$, $T_{Fi}$, $T_{Fo}$, $T_{Fb}$ represents the early fault datasets for the cage rubbing, bearing wear, inner ring, outer ring, and surface damage of the balls, respectively. Finally, *D* is randomly divided into the training set, validation set, and test set, namely, $D_{train}$, $D_{validation}$, and $D_{test}$.

This paper aims to construct a multi-classifier *f* for the operation status of the space wheel rotor system based on $D_{train}$ and $D_{validation}$, to determine whether there is a fault in the system (binary classification) and to locate the fault (multi-classification) while satisfying the small number of ntrain ($ntrain_{F}$<$ntrain_{N}$) and whether there is a lack of fault types in the training set. Based on the self-learning characteristics of the mode, the data in the validation set are relabeled, and the model is retrained. The diagnostic results of the model are validated by $D_{test}$.

## 3. Methods

### 3.1. Overview

To solve the multi-classification problem of the early fault diagnosis of the space wheel rotor system in the case of insufficient labeled data and missing fault types, we propose an intelligent fault diagnosis method based on the HB-SC-MCCNN model. Figure 1 shows the overview of the proposed method.

The process of the method is as follows:Step 1: Multi-source signals acquisition. Multi-source signals include the operation trajectory of the cage, dynamic friction torque, and vibration. By utilizing complementary information from different data sources at the same time, the complete and consistent information description of the rotor system can be effectively improved, making fault multi-classification more accurate and reliable.Step 2: Dataset dividing. The original vibration signals are directly divided into the training set, verification set, and test set without any signal processing or feature extraction.Step 3: Model training. The HB-SC-MCCNN model proposed in this paper is trained by training samples and provides a relabeling function for the validation set data after preliminary training, and is then retrained.Step 4: Intelligent early fault diagnosis. The trained model in Step 3 analyzes the data in the test set, determines whether there is a fault in the rotor system, and locates the type of fault.

### 3.2. HB-SC-MCCNN

The model architecture of HB-SC-MCCNN is shown in Figure 2. This model directly takes original multi-source data as input and predicts the early faults through the hierarchical mechanism of multiple branch output layers, which outputs the status and fault location of the space wheel rotor system simultaneously.

#### 3.2.1. Stratification of the Rotor System

According to the actual diagnostic requirements, we divide the faults into three levels: fault detection (a binary classification problem, which determines whether the rotor system has a fault), fault evaluation (a multi-classification problem, which determines the type of fault in the rotor system), and fault location (a multiple classification problem, which determines the specific part where the fault occurs). The hierarchical structure is shown in Table 1 (we use “level” to represent different levels in a hierarchical structure, and “layer” to represent layers in a neural network). By dividing the data categories hierarchically, we can limit the errors that may occur in diagnosis to a subcategory. For example, the model may not be able to distinguish whether the fault occurred in the inner or outer ring, but it can distinguish whether there is a fault in the rotor system.

#### 3.2.2. Basic Structural Block

We integrate similarity clustering into a parameter-shared dual MCCNN architecture (as shown in Figure 3 and Tabel Table 2) and use it as the basic structural block of HB-SC-MCCNN.

Unlike the Softmax function, which is connected after the full connection layer to the output classification labels in traditional convolutional networks, the SC adopted in this paper is used to cluster the relevant features extracted from MCCNN through similarity measures. SC not only effectively reduces the dependence on the amount of label data during classification but also utilizes the characteristics of SC comparison clustering to enable convolutional neural networks to have the self-learning ability for new fault types.

It is worth noting that to ensure that the same type of data has better relevant characteristics in the target space, we establish a dual MCCNN architecture with parameter sharing. In a single iteration, each channel of the MCCNN input data belongs to the same type.

The process of the method is as follows:

**Step 1:** Relevant characteristics. An MCCNN with input channel number $c_{i}$ and output channel number $c_{o}$ is constructed ($c_{i}>c_{o}>1$) [35]. The kernel arrays assigned to each input channel with the shape of $k_{h}\times k_{w}$ are concatenated on input channel dimension to obtain the convolutional kernel with the shape of $c_{i}\times k_{h}\times k_{w}$. The input multi-channel samples and convolutional kernel are subjected to correlation operations to obtain the two-dimensional cross-correlation features.

It is worth noting that it is not feasible to replace MCCNN with the single-channel CNN [36,37], which is commonly used in current intelligent fault diagnosis methods. If the characteristics of the input data in the target space do not correlate, the similarity clustering effect will be affected, especially in the case of the number of training samples being insufficient.

**Step 2:** Similarity measure. $E\left({\bar{x}}_{1},{\bar{x}}_{2}\right)$ can be considered an “energy” function [38], used to measure the relevant features ${\bar{x}}_{1}$ and ${\bar{x}}_{2}$ extracted by MCCNN and the similarity between them, specifically defined as
(1)$$E\left({\bar{x}}_{1},{\bar{x}}_{2}\right)={∥{\bar{x}}_{1}-{\bar{x}}_{2}∥}_{1}={∥G\left(x_{1}\right)-G\left(x_{2}\right)∥}_{1},$$

$G\left(x\right)$ in the above equation is the one-dimensional correlation feature vector output after the input sample enters the convolutional layer and fully connected layer of MCCNN.

It is worth noting that it is not feasible to replace the $L_{1}$ norm with a square norm for $E\left({\bar{x}}_{1},{\bar{x}}_{2}\right)$. If $E\left({\bar{x}}_{1},{\bar{x}}_{2}\right)$ is the square norm of the difference between ${\bar{x}}_{1}$ and ${\bar{x}}_{2}$, when $E\left({\bar{x}}_{1},{\bar{x}}_{2}\right)$ approaches 0, the gradient of $E\left({\bar{x}}_{1},{\bar{x}}_{2}\right)$ relative to the model parameters will disappear [39].

**Step 3:** Parameter optimization. The loss function of the model is defined as
(2)$$\ell \left(\theta \right)=\sum_{i=1}^{n_{train}}L\left(\theta ,{\left(Y,x_{1},x_{2}\right)}^{i}\right),\hat{\theta }=\underset{\theta }{argmin}\ell \left(\theta \right),$$
where ${\left(Y,x_{1},x_{2}\right)}^{i}$ is the *i*-th sample, consisting of input data $x_{1},x_{2}$ and the label *Y*. The *Y* in Equation (Equation 2) is a binary label (0 or 1). When $x_{1}$ and $x_{2}$ belong to the same type, Y=1. When $x_{1}$ and $x_{2}$ belong to different types, Y=0, and $x_{2}$ is represented as $x_{2}^{{}^{\prime }}$. In supervised and semi-supervised learning, the label of the training data is known, so the optimal value of label *Y* can be determined. $\hat{\theta }$ is the optimized result, and $\underset{\theta }{argmin}\ell \left(\theta \right)$ represents an operational process, which involves selecting appropriate model parameters θ to make $\ell \left(\theta \right)$ reach the minimum.

When uniting Equations (1) and (2), the loss function could be indirectly related to the input data and model parameters through $E\left({\bar{x}}_{1},{\bar{x}}_{2}\right)$:(3)$$\ell \left(\theta \right)=\sum_{i=1}^{n_{train}}\left[Y\ell_{s}\left(E{\left({\bar{x}}_{1},{\bar{x}}_{2}\right)}^{i}\right)-\left(1-Y\right)\ell_{d}\left(E{\left({\bar{x}}_{1},{\bar{x}}_{2}^{{}^{\prime }}\right)}^{i}\right)\right],$$
where $\ell_{s}$ represents the partial loss function of the same type, while $\ell_{d}$ represents the partial loss function of the different types.

$\ell_{s}$ and $\ell_{d}$ are designed in the principle that the minimization of l will decrease the energy of the same type and increase the energy of the different type. A simple way to achieve that is to make $\ell_{s}$ monotonically increasing and $\ell_{d}$ monotonically decreasing.

To meet the more general conditions and ensure that the same type of data and different types of data always have the boundary $E\left({\bar{x}}_{1},{\bar{x}}_{2}\right)+m<E\left({\bar{x}}_{1},{\bar{x}}_{2}^{{}^{\prime }}\right)$ (where the integer *m* is the boundary), the exact loss function is expressed as
(4)$$\ell \left(\theta \right)=\sum_{i=1}^{n_{train}}\left[Y\frac{2}{Q}{\left(E{\left({\bar{x}}_{1},{\bar{x}}_{2}\right)}^{i}\right)}^{2}-\left(1-Y\right)2Qexp\left(-\frac{2.77}{Q}{\left(E\left(\bar{x_{1}},{\bar{x}}_{2}^{{}^{\prime }}\right)\right)}^{i}\right)\right],$$
where the constant *Q* is the upper boundary of $E\left({\bar{x}}_{1},{\bar{x}}_{2}\right)$, which is determined by the one-dimensional correlated feature vectors $G\left(x\right)$ output by TCCNN, according to Equation (Equation 1).

It is worth noting that there must be a contrastive term in the loss function of the model to ensure that the “energy” from the same type of data is low, while the “energy” from different types of data is high. Once the contrastive term disappears, the energy and the loss can be made zero by simply making $G\left(x_{1}\right)$ a constant function [40,41].

The *Y* in Equation (Equation 2) is a binary label (0 or 1). When ${\bar{x}}_{1}$ and ${\bar{x}}_{2}$ belong to the same type, Y=1. When ${\bar{x}}_{1}$ and ${\bar{x}}_{2}$ belong to the different type, Y=0, and ${\bar{x}}_{2}$ is represented as ${\bar{x}}_{2}^{{}^{\prime }}$.

It is worth noting that in supervised and semi-supervised learning, the label of training data is known to the user, so the optimal value of label *Y* can be determined.

Then, we could optimize model parameters θ through the small-batch random gradient descent algorithm [42] as Equation (Equation 3):(5)$$\hat{\theta }\leftarrow \theta -\frac{\eta }{\left|B\right|}\sum_{B}^{}\left(\frac{\partial \ell_{s}}{\partial \theta }-\frac{\partial \ell_{d}}{\partial \theta }\right),$$
where η>0 is the learning rate, and *B* is the batch size.

#### 3.2.3. Branch Structure

The introduction of a branch structure perfectly integrates the hierarchical structure of MCCNN network features [43] with the natural classification level of the fault data of the space wheel rotor system [44] and diagnoses faults at different classification levels.

The hierarchical structure of MCCNN network features refers to the fact that the lower layers of MCCNN typically capture low-level features, while the higher layers can extract high-level features. Therefore, each layer of MCCNN contains a hierarchical structure of network features.

The natural classification hierarchy of the rotor system fault data could be explained by taking the established early fault dataset of the satellite flywheel rotor system as an example. In this benchmark dataset, the normal and fault states of the system are easy to distinguish, but it is difficult to recognize the fault of the inner ring, outer ring, or balls’ surface. The reason is that normal and faulty system vibration signals belong to different rough categories, while vibration signals of different fault types belong to the same rough category. For the convenience of expression, only vibration signals are used as an example here. The normal bearing vibration signal belongs to irregular random vibration. However, when a surface damage-like defect occurs in a bearing of the rotor system, if the damage point rolls over the surface of the bearing, sudden shock vibrations will occur, which results in periodic shock pulses in the vibration signal.

Compared to the traditional multi-classification method (as shown in Figure 4), HB-SC-MCCNN simplifies the multi-classification problem into a multi-step binary classification problem by introducing the branch structure, which effectively reduces the dependence on the amount of labeled data when directly performing multi-classification as shown in Figure 5.

## 4. Experiments

### 4.1. Description of the Dataset

The dataset adopted for experiments is obtained from the established visual multi-performance monitoring rig for the space flywheel rotor system, which is mainly composed of four parts—rotating driver, signal collector, A/D converter, and upper computer—as shown in Figure 6. And the signal collector mainly includes a high-speed camera, a dynamic friction torque instrument, and a vibration sensor. Furthermore, the partial parameters of the used sensors are shown in Table 3.

Figure 7 shows the structure of the rotor system as the experimental object. And Table 4 lists the main parameters of the testing bearing modeled B7004C that works with a speed of 6000 r/min and an axial load of 100 N. The marking points distribute concentrically and uniformly on the bearing cage capture and track marks of the running trajectory of the cage used by the high-speed camera systems. The sampling frequency of the experiments is 10 kHz, with 47,690 sampling points. It can be seen that the samples in the dataset are all early faults of the rotor system, that is, real faults generated during ground development and testing.

The normal states of the cage and bearing raceway are shown in Figure 8 and Figure 9, with no abnormal contact or rubbing marks on the outer surface of the cage, and uniform contact points in the cage pockets. In addition, the contact area between the inner and outer rings of the bearing and balls during operation is normal, and no defects such as wear and surface damage can be observed there.

Figure 10 shows the early fault of cage rubbing. The abnormal rub between the outer diameter surface of the porous oil-containing cage and the guide surface of the outer ring caused shear deformation and fracture at the edge of the pores, which is manifested as the carbonization and blackening of polyimide materials. And the abnormal friction and relative sliding between pockets and balls also cause excessive blackening and scratches in the contact area.

The wear of the bearings can be seen in Figure 11, where the material transfer occurs in the form of particle shedding in the operating contact area of the rings and balls, causing the machining marks on the original machining surface to be worn, and manifested as a “whitening” phenomenon in the contact area. The surface damage of the inner ring, outer ring, and balls is shown in Figure 12, with surface damage such as indentation and scratches occurring in the contact area during rotation. Table 5 lists the data information of the established early fault dataset $D_{1}$ of the space flywheel rotor system. Samples in the dataset represent the segmented signals with 1024 consecutive data points randomly selected from the original multi-source signals shown in Figure 13.

### 4.2. Results

The experiment used 28 datasets to verify the effectiveness and adaptability of the proposed HB-SC-MCCNN. $D_{2}$ to $D_{28}$ were constructed based on $D_{1}$ as shown in Table 6.

The number of labeled samples in $D_{1}$ to $D_{4}$ decreases gradually, while the number in $D_{5}$ to $D_{24}$ also decreases gradually, coexisting with the missing of 1/5 fault types.At the same time, $D_{25}$ to $D_{27}$ miss 2/5, 3/5, and 4/5 fault types, and the fault data are completely missing in $D_{28}$.

The reduced data samples from $D_{2}$ to $D_{28}$ were placed in the validation set, and relabeled during the training process. The relabeled samples were combined with the original labeling samples to retrain the model. Moreover, to evaluate the performance of the model under harsh conditions, the number of samples is kept constant in the test set, the number of normal samples (*N*) in the test set is 1100, and the number of early fault samples of $F_{c}$, $F_{w}$, $F_{i}$, $F_{o}$, and $F_{b}$ is 220.

For avoiding accidental test results, the 10-fold cross-validation method was adopted, and the average and standard deviations of the multiple verification results of the test were calculated. The results obtained from the test results (as shown in Table 6) are as follows.

The accuracy of the test results in the $D_{1}$ to $D_{4}$ datasets is above 99.98%, and there is no significant trend of change as the number of labeled samples decreases. The test results show that the HB-SC-MCCNN model proposed in this paper could still have the excellent diagnostic ability with a small number of samples. In addition, the confusion matrix of the model’s prediction results on datasets $D_{1}$ to $D_{4}$ is shown in Figure 14.

The accuracy of the test results for the $D_{5}$ to $D_{24}$ datasets is above 99.87%, and there is no significant trend of change as the number of labeled samples decreases in the absence of 1/5 fault types. The test results show that the proposed model still has excellent diagnostic results under the condition of a small number of samples and 1/5 types of missing faults.

The accuracy of the test results for $D_{25}$ is 84.93%, and the accuracy of the first-level binary classification is 100% (this result is not indicated in Table 4). The test results indicate that the proposed model still has good fault location results when 2/5 fault types are missing, and could effectively ensure the accuracy of the fault detection.

The accuracy of the test results for $D_{26}$ and $D_{27}$ is 73.79% and 63.31%, respectively, and the accuracy of the first-level binary classification is 100% and 99.87%. The test results show that the proposed model loses the accuracy of the fault location when 2/5 or 3/5 fault types are missing but can still effectively ensure the accuracy of fault detection.

The accuracy of the test results for the $D_{28}$ dataset is only 25%, and the accuracy of the first-level binary classification is 50%, from which it can be determined that HB-SC-MCCNN has failed in the case of the complete missing of fault samples in the training set.

### 4.3. Ablation Experiment

#### 4.3.1. Comparative Experiment for Exploring the Contribution of MCCNN to HB-SC-MCCNN

The comparative experiment adopts CNN to replace MCCNN, and the model keeps the same except for $c_{i}$ and $c_{o}$. The comparative results (shown in Table 7 and Figure 15) show that the accuracy of the HB-SC-CNN and HB-SC-MCCNN test results on the datasets decreased from 99.87% to 90.47%, and the standard deviation increased from 0.15 to 9.12, which indicated the increase in the dispersion of the test results. With the decrease in the sample number, the test accuracy of HB-SC-CNN presents a slight downward trend.

We can conclude that the relevant features extracted by MCCNN in the target space effectively improve the clustering effect of SC.

#### 4.3.2. Comparative Experiment for Exploring the Contribution of HB and SC to HB-SC-MCCNN

CNN with softmax is the most traditional and mature deep learning model, which uses CNN to extract the features of the input data, and classifies them through softmax. However, this model lacks the self-learning ability of new fault types. When there are missing fault types in the training set, relabeling and model retraining would not be performed on the samples in the validation set. Comparing the experiment results (as shown in Table 7 and Figure 15), it can be seen that the accuracy of the test results of softmax-CNN and HB-SC-MCCNN on the $D_{24}$ dataset decreased from 99.87% to 72.70%, and the standard deviation increased from 0.15 to 18.85. With the decrease in the number of samples in the training dataset, the test accuracy of softmax-CNN is decreased significantly, which is also consistent with the conclusion that the current commonly used model [19,20,40] would have a significant decrease in model performance as the number of samples decreases.

### 4.4. Comparison with Existing Models

In existing research, there are few fault diagnosis methods for rotating machines that simultaneously focus on insufficient labeled samples and missing fault types. The deep representation clustering-based fault diagnosis method with unsupervised data proposed by Li et al. [27] and Zhao et al. [28] achieved excellent diagnostic results. The test results (as shown in Table 7 and Figure 15) indicate that in the case of sufficient samples and 1/5 fault types missing ($D_{5}$ to $D_{9}$), the accuracy of the deep representation clustering is consistent with that of the HB-SC-MCCNN, and is superior to softmax-CNN. Automatic encoders and distance metric learning play important roles in the deep representation clustering, which enables the model to have the self-learning ability for new fault types.

In the case of insufficient training set samples and missing 1/5 fault types ($D_{20}$ to $D_{24}$), compared with HB-SC-MCCNN, the test accuracy of the deep representation clustering decreases slightly, and the standard deviation increases slightly. Test results on $D_{24}$ are decreased from 99.87% to 92.58%, and the standard deviation is increased from 0.15 to 6.55. The reason for this phenomenon is that the deep representation clustering still uses the traditional direct multi-classification method, which relies more on the amount of labeled data than the multi-step two-classification method used in this article, and is also consistent with the analysis in Part 3.2.3.

In the absence of 2/5 fault types ($D_{25}$), the test accuracy of deep representation clustering is decreased by 27.89% compared to HB-SC-MCCNN, and the standard deviation is increased by 9.01. The k-means clustering method used by deep representation clustering has a negative impact on the results, which cannot ensure the accuracy of relabeling in the absence of two or more fault types. However, the similarity clustering method used in this paper is effective for making the proposed model more competitive.

## 5. Conclusions

Aiming at the actual needs of fault diagnosis of the space wheel rotor system, this paper divides the fault into three levels: fault detection (a binary classification problem, which determines whether the rotor system has a fault), fault evaluation (a multi-classification problem, which determines the type of fault in the rotor system), and fault location (a multiple classification problem, which determines the specific part where the fault occurs). At the same time, considering the frequent problem of insufficient labeled samples and missing fault types in practical applications, an intelligent fault diagnosis method for the space wheel rotor system based on HB-SC-MCCNN is proposed.

The model uses SC-MCCNN as the basic structural block and simultaneously achieves fault detection and network self-learning capabilities when the number of labeled samples is insufficient. A hierarchical branch structure (HB) is introduced into the model, and the multi-classification problem is simplified to a multi-step binary classification problem, which further reduces the dependence on the number of labeled samples when locating the fault.

Through relabeling unlabeled data and fault missing types, as well as retraining the network model, the proposed method has achieved excellent diagnostic results. Experimental results show that the diagnostic accuracy of the proposed method is above 99.87% when 1/5 fault types are missing, and there is no obvious trend of change as the number of labeled samples decreases, which is more competitive compared with existing models.

However, although the proposed HB-SC-MCCNN model exhibits excellent diagnostic performance and adaptability, it still loses the accuracy of fault location when 2/5 or 3/5 fault types are missing. More efficient basic structural blocks should be further explored to reduce the computing costs. An exploration of more efficient multi-classification simplification methods should also be included in further research directions.

## Figures and Tables

**Figure 1 sensors-23-08198-f001:**
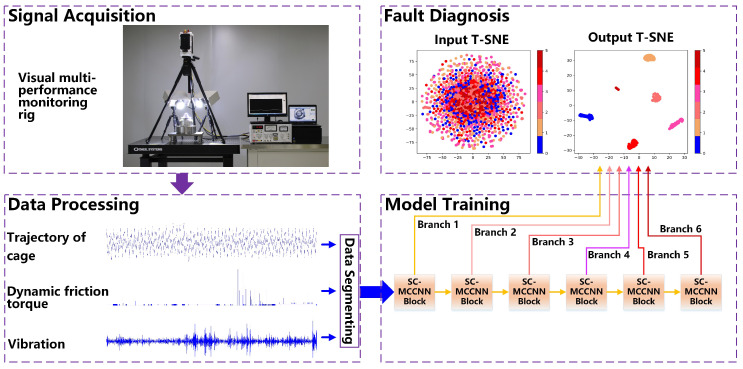
Overview of the proposed method.

**Figure 2 sensors-23-08198-f002:**
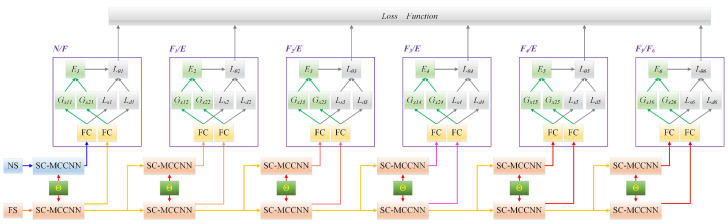
Architecture of HB-SC-MCCNN.

**Figure 3 sensors-23-08198-f003:**
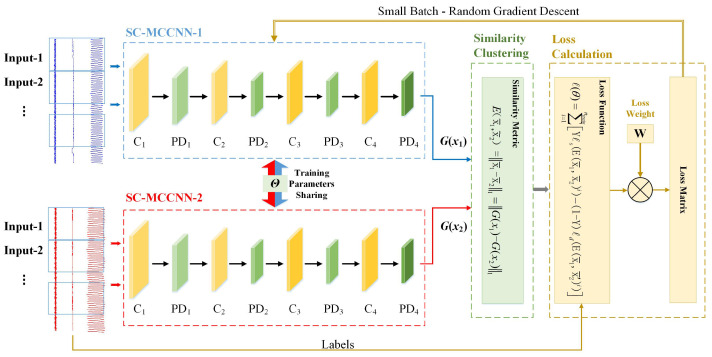
The basic structural block of proposed model: MCCNN block.

**Figure 4 sensors-23-08198-f004:**
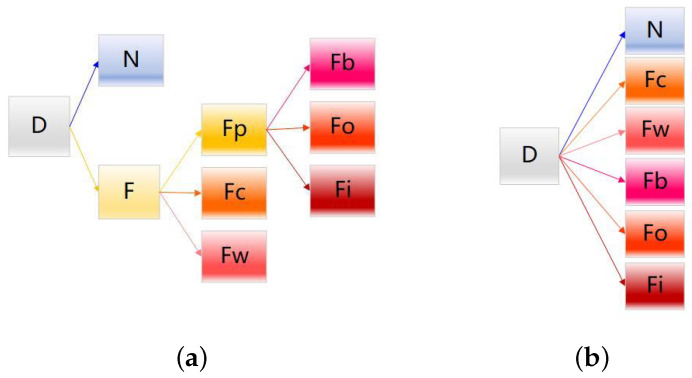
Traditional multi-classification methods.

**Figure 5 sensors-23-08198-f005:**
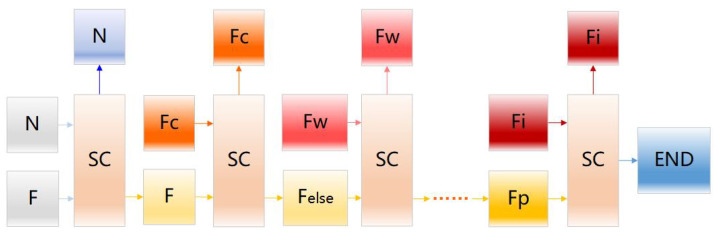
The proposed multi-step binary classification.

**Figure 6 sensors-23-08198-f006:**
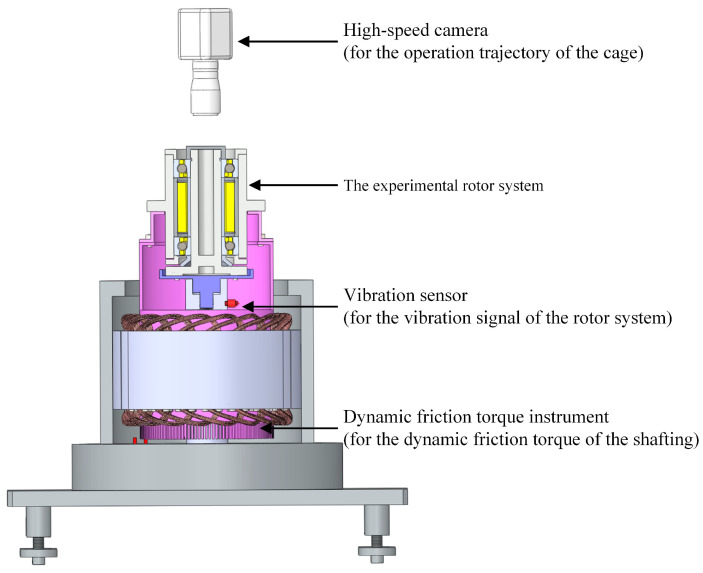
The visual multi-performance monitoring rig for space flywheel rotor system.

**Figure 7 sensors-23-08198-f007:**
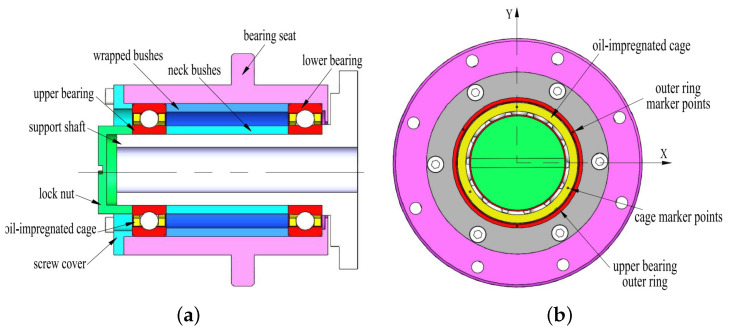
Structure of the experimental object: space wheel rotor system.

**Figure 8 sensors-23-08198-f008:**
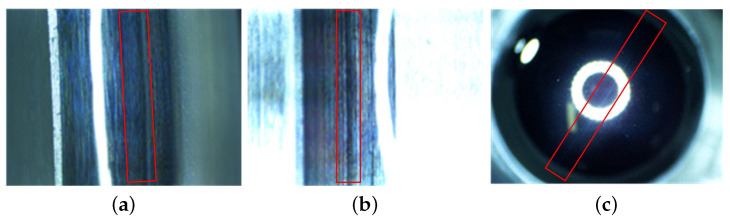
Structure of the experimental object: space wheel rotor system. The red box represents the rotating contact area. (**a**) Other ring; (**b**) inner ring; (**c**) ball.

**Figure 9 sensors-23-08198-f009:**
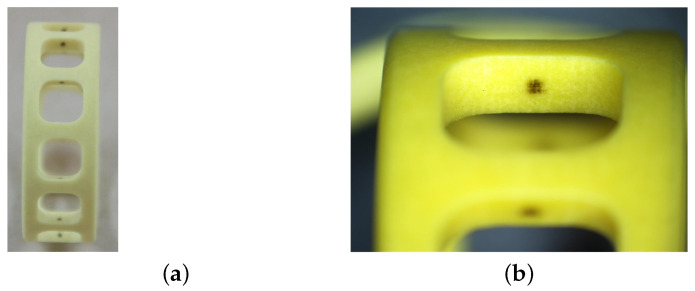
Normal states of the cage. (**a**) The outer surface; (**b**) the pockets.

**Figure 10 sensors-23-08198-f010:**
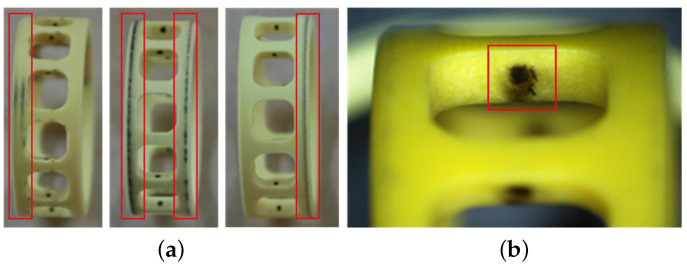
The early fault of cage rubbing. The red box represents the fault area. (**a**) The outer surface; (**b**) the pockets.

**Figure 11 sensors-23-08198-f011:**
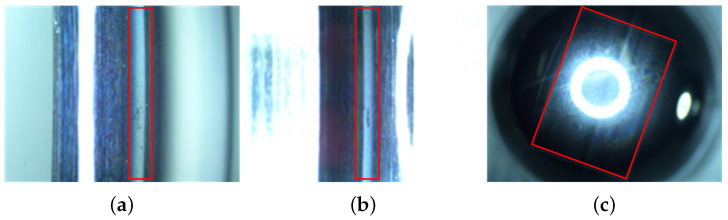
The wear of bearings. (The red box represents the fault area. (**a**) Other ring; (**b**) inner ring; (**c**) ball.

**Figure 12 sensors-23-08198-f012:**
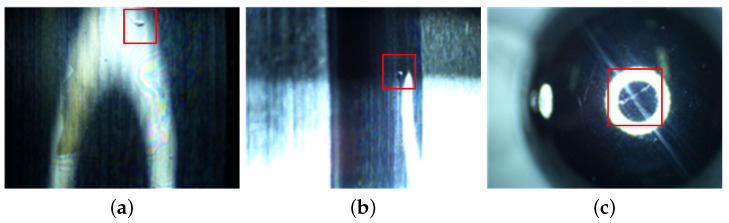
The surface damage of bearings. The red box represents the fault area. (**a**) Other ring; (**b**) inner ring; (**c**) ball.

**Figure 13 sensors-23-08198-f013:**
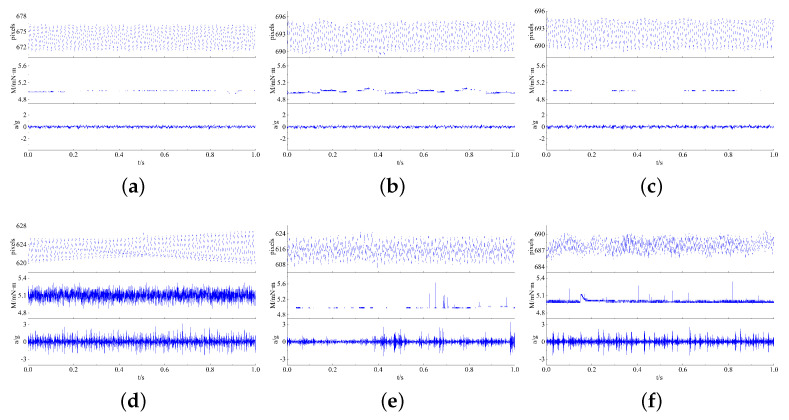
The segmented signals randomly selected from the original multi-source signals of the space flywheel rotor system. (**a**–**c**) Normal signals; (**d**–**f**) early fault signals.

**Figure 14 sensors-23-08198-f014:**
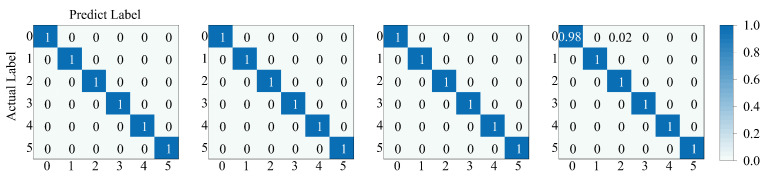
Confusion matrix of the model’s prediction results on datasets $D_{1}$ to $D_{4}$.

**Figure 15 sensors-23-08198-f015:**
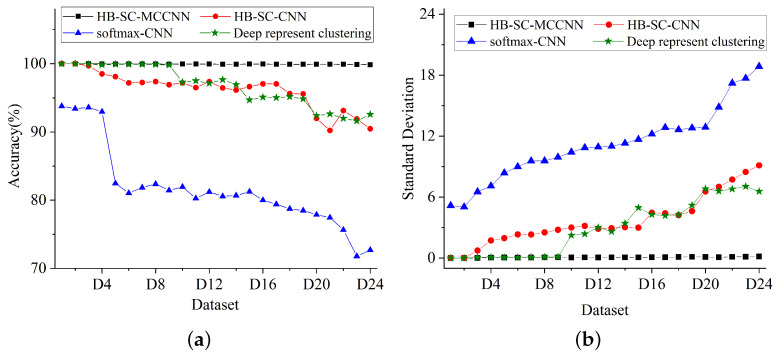
Results of the comparative experiments. (**a**) Accuracy; (**b**) standard deviation.

**Table 1 sensors-23-08198-t001:** Stratification of the fault in rotor system.

Level #1	Level #2	Level #3
Normal	Normal	
	Cage rubbing	
	Bearing wear	
Falut		Inner ring
	Damage on the bearing surface	Outer ring
		Balls

**Table 2 sensors-23-08198-t002:** Detailed parameters of MCCNN.

Layers	Kernel Size/Stride	Output	Activation Function	Padding in Pooling Layer	Dropout Rate	Trainable Parameters
Input	/	2, 1024 × 1	/	/	/	0
C1	(2, 32, 11 × 1)	32, 1024 × 1	ReLU	/	/	12, 662
PD1	(2 × 1)	32, 512 × 1	/	0-padding	0.5	0
C2	(32, 64, 11 × 1)	64, 512 × 1	ReLU	/	/	67, 712
PD2	(2 × 1)	64, 256 × 1	/	0-padding	0.5	0
C3	(64, 128, 11 × 1)	128, 256 × 1	ReLU	/	/	196, 864
PD3	(2 × 1)	128, 128 × 1	/	0-padding	0.5	0
C4	(128, 256, 11 × 1)	256, 128 × 1	ReLU	/	/	786, 944
PD4	(1 × 2)	128, 128, 1	/	0-padding	0.5	0
F	128	128 × 1	ReLU	/	/	33, 154

**Table 3 sensors-23-08198-t003:** Parameters of the sensors used in the visual multi-performance monitoring rig.

Sensor 1, High-Speed Camera	
Manufacturer	Vision Research
Model	Phantom V711
Full frame shooting speed [frames/s]	7530
Maximum shooting speed [frames/s]	1.4 million
Maximum resolution	1280×800
Pixel size [µm]	20
**Sensor 2, Friction Torque Measuring Instrument**	
Manufacturer	China Luoyang Bearing Research Institute Co., Ltd.
Model	ZYS−2
Test object	Bearings/Shafting
Range [mN· m]	0∼100
Speed [r/min]	1∼9990
Error	<3
**Sensor 3, Vibration Sensor**	
Manufacturer	Chengdu Hangzhen Automation Equipment Co., Ltd., China
Model	YD−1
Charge Sensitivity	6∼10 pC/m/s ${}^{2}$
Response frequency	1∼10,000 Hz

**Table 4 sensors-23-08198-t004:** Stratification of the fault in rotor system.

Parameter	Value
Diameter of outer ring [mm]	42
Diameter of inner ring [mm]	20
Width of rings [mm]	12
Diameter of balls [mm]	6.35
Number of steel balls	12
Outer diameter of cage [mm]	34.1
Inner diameter of cage [mm]	28.5
Width of cage [mm]	11.4
Shape of pockets	Square
Guide surface of cage	Outer surface
Contact angle [deg]	15
Material of rings	G95Cr18
Material of balls	G95Cr18
Material of cage	Polyimide

**Table 5 sensors-23-08198-t005:** Stratification of the fault in rotor system.

Level #1	Level #2	Level #3	Number of Samples	
Normal	Normal		5500	5500
	Cage rubbing		1100	1100
	Bearing wear		1100	1100
Fault	Damage on the bearing surface	Inner ring	1100	3300
		Outer ring	1100	0
		Balls	1100	0

**Table 6 sensors-23-08198-t006:** Dataset information and corresponding experimental results.

Data Sets	Training Sets (Labeled Samples)		Validation Sets (Labeled Samples)		Accuracy (%)
	N	$F_{c}/F_{w}/F_{i}/F_{o}/F_{b}$	N	$F_{c}/F_{w}/F_{i}/F_{o}/F_{b}$	
$D_{1}$	3300	660/660/660/660/660	1100	220/220/220/220/220	100 (0)
$D_{2}$	1650	330/330/330/330/330	2750	550/550/550/550/550	100 (0)
$D_{3}$	825	165/165/165/165/165	3575	715/715/715/715/715	100 (0)
$D_{4}$	400	80/80/80/80/80	4000	800/800/800/800/800	99.98 (0.04)
$D_{5}$	2640	660/660/660/660/0	1760	220/220/220/220/880	99.96 (0.05)
$D_{6}$	2640	660/660/660/0/660	1760	220/220/220/880/220	99.97 (0.04)
$D_{7}$	2640	660/660/0/660/660	1760	220/220/880/220/220	99.96 (0.05)
$D_{8}$	2640	660/0/660/660/660	1760	220/880/220/220/220	99.95 (0.06)
$D_{9}$	2640	0/660/660/660/660	1760	880/220/220/220/220	99.94 (0.06)
$D_{10}$	1320	330/330/330/330/0	3080	550/550/550/550/880	99.96 (0.05)
$D_{11}$	1320	330/330/330/0/330	3080	550/550/550/880/550	99.96 (0.07)
$D_{12}$	1320	330/330/0/330/330	3080	550/550/880/550/550	99.95 (0.05)
$D_{13}$	1320	330/0/330/330/330	3080	550/880/550/550/550	99.94 (0.07)
$D_{14}$	1320	0/330/330/330/330	3080	880/550/550/550/550	99.93 (0.07)
$D_{15}$	660	165/165/165/165/0	3740	715/715/715/715/880	99.96 (0.07)
$D_{16}$	660	165/165/165/0/165	3740	715/715/715/880/715	99.95 (0.08)
$D_{17}$	660	165/165/0/165/165	3740	715/715/880/715/715	99.94 (0.08)
$D_{18}$	660	165/0/165/165/165	3740	715/880/715/715/715	99.92 (0.10)
$D_{19}$	660	0/165/165/165/165	3740	880/715/715/715/715	99.92 (0.11)
$D_{20}$	320	80/80/80/80/0	4080	800/800/800/800/880	99.93 (0.10)
$D_{21}$	320	80/80/80/0/80	4080	800/800/800/880/800	99.93 (0.08)
$D_{22}$	320	80/80/0/80/80	4080	800/800/880/800/800	99.92 (0.11)
$D_{23}$	320	80/0/80/80/80	4080	800/880/800/800/800	99.91 (0.13)
$D_{24}$	320	0/80/80/80/80	4080	880/800/800/800/800	99.87 (0.15)
$D_{25}$	3300	660/660/660/0/0	1100	220/220/220/880/880	84.93 (8.29)
$D_{26}$	3300	660/660/0/0/0	1100	220/220/880/880/880	73.79 (10.37)
$D_{27}$	3300	660/0/0/0/0	1100	220/880/880/880/880	63.31 (11.94)
$D_{28}$	3300	0/0/0/0/0	1100	880/880/880/880/880	25 (0)

**Table 7 sensors-23-08198-t007:** Results of the comparative experiments.

Datasets	HB-SC-MCCNN	HB-SC-CNN	Softmax-CNN	Deep Represent Clustering
$D_{1}$	100 (0)	100 (0)	93.78 (5.15)	100 (0)
$D_{2}$	100 (0)	100 (0)	93.42 (5.03)	100 (0)
$D_{3}$	100 (0)	99.70 (0.74)	93.60 (6.52)	99.94 (0.05)
$D_{4}$	99.98 (0.04)	98.51 (1.73)	92.97 (7.08)	99.92 (0.09)
$D_{5}$	99.96 (0.05)	98.10 (2.08)	82.47 (8.98)	99.92 (0.04)
$D_{6}$	99.97 (0.04)	97.19 (2.45)	81.04 (8.94)	99.92 (0.05)
$D_{7}$	99.96 (0.05)	97.25 (2.54)	81.85 (9.55)	99.91 (0.07)
$D_{8}$	99.95 (0.06)	97.38 (2.73)	82.37 (9.57)	99.89 (0.09)
$D_{9}$	99.94 (0.06)	96.92 (2.89)	81.43 (9.92)	99.82 (0.11)
$D_{10}$	99.96 (0.05)	97.20 (3.07)	81.95 (10.42)	97.30 (2.24)
$D_{11}$	99.96 (0.07)	96.50 (3.09)	80.27 (10.86)	97.50 (2.37)
$D_{12}$	99.95 (0.05)	97.37 (4.41)	81.21 (10.93)	97.13 (3.01)
$D_{13}$	99.94 (0.07)	96.46 (4.79)	80.56 (11.00)	97.67 (2.61)
$D_{14}$	99.93 (0.07)	96.15 (6.44)	80.68 (11.30)	96.93 (3.41)
$D_{15}$	99.96 (0.07)	96.64 (7.86)	81.24 (11.68)	94.70 (4.96)
$D_{16}$	99.95 (0.08)	97.05 (8.52)	80.01 (12.21)	95.01 (4.28)
$D_{17}$	99.94 (0.08)	97.05 (4.41)	79.40 (12.84)	95.03 (4.17)
$D_{18}$	99.92 (0.10)	95.62 (4.22)	78.74 (12.64)	95.16 (4.26)
$D_{19}$	99.92 (0.11)	95.58 (4.61)	78.48 (12.79)	94.85 (5.19)
$D_{20}$	99.93 (0.10)	91.99 (6.55)	77.88 (12.87)	92.40 (6.81)
$D_{21}$	99.93 (0.08)	90.23 (7.00)	77.45 (14.85)	92.63 (6.58)
$D_{22}$	99.92 (0.11)	93.14 (7.71)	75.64 (17.21)	91.99 (6.80)
$D_{23}$	99.91 (0.13)	91.90 (8.46)	71.79 (17.69)	91.62 (7.04)
$D_{24}$	99.87 (0.15)	90.47 (9.12)	72.70 (18.85)	92.58 (6.55)
$D_{25}$	84.93 (8.29)	77.15 (15.08)	47.52 (20.65)	57.04 (9.16)
$D_{26}$	73.79 (10.37)	66.12 (17.87)	45.7 (22.09)	50.98 (13.55)
$D_{27}$	63.31 (11.94)	54.01 (19.50)	39.62 (24.13)	48.05 (14.98)
$D_{28}$	25 (0)	25 (0)	25 (0)	25 (0)

## Data Availability

Data available on request due to restrictions, e.g., privacy or ethical. The data presented in this study are available on request from the corresponding author. The data are not publicly available due to the confidentiality policy.

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
