# Peer review of "Intelligent Early Fault Diagnosis of Space Flywheel Rotor System"

_sensors, 2023, doi:10.3390/s23198198_

Round 1

Reviewer 1 Report

Dear authors,
This article addresses an interesting topic of fault diagnosis of space flywheel rotor system. After careful reading of the submitted manuscript, I have pointed out areas that require the addition of knowledge or improvement of the content.
1.    In line 110 Multi-source signals acquisition was used. Please provide used sensors product, therefore reader can have access to datasheets of the operation trajectory of the cage, dynamic friction torque, and vibration.
2.    How was vibration sensor aligned. Was it 1-axis sensor or 3-axis sensor?
3.    3. Faults was divided into three levels: fault detection, fault evaluation, and fault location which were treated as binary classification problem. What features of signal were used,  raw signal, time-domain, frequency domain or time-frequency domain.
4.    For the convenience of expression, only vibration signals are used as an example here. What dominant frequencies appears in vibration signal?
5.    Line 247. Segmented signals with 1024 consecutive data points were shown. What is time length of analysed time-window signal (1024/10kHz = 0.1024 s ?), why this number of samples, how many channels were used and what was the overlap?
6.    Fig 13 labels are not readable.
7.    Please provide representative confusion matrix.
8.    The state-of-the-art of fault diagnosis of electromechanical rotary system must be extended. Extend references to other methods in the field of fault diagnosis of electromechanical rotary system from last years e.g. https://doi.org/10.3390/s23073755 https://doi.org/10.3390/electronics12163460 https://doi.org/10.17531/ein/170114  https://doi.org/10.3390/act12030125 https://doi.org/10.1109/TII.2022.3147828 .
9.    Underline the benefits and drawbacks of the proposed method compared to existing ones.

Reviewer 2 Report

This paper proposes an intelligent early fault diagnosis method based on the multi-channel convolutional neural network with hierarchical branch and similarity clustering (HB-SC-MCCNN). Results of the experiments for comparing the abilities between the proposed method and several advanced deep learning models confirm that on the established early fault dataset of the space flywheel rotor system, the proposed method successfully achieves the hierarchical diagnosis, and presents stronger competitiveness in the case of insufficient labeled data and missing fault types at the same time.

The manuscript can be accepted after minor modifications.

1、The section 2 Problem formulation is unnecessary, this paper is a classical multi-class problem. There is nothing special;

2、The author should pay attention to the clarity and format of the pictures in the text;

3、Eq.2 requires specific mathematical representations of ls and ld for a more intuitive understanding. The physical meaning of the other symbols in this equation also should be supplemented.

4、How do the authors set up the network structure and parameters of the model in Fig.3? It is suggested to list a table

5、The scientific writing and presentation skills of this article should be improved.

 Minor editing of English language required
